# Measurement-induced phase transition in interacting bosons from most likely quantum trajectory

**Anna Delmonte[1*], Zejian Li[2], Rosario Fazio[2,3] and Alessandro Romito[4]**

**1** SISSA, Via Bonomea 265, I-34136 Trieste, Italy

**2** The Abdus Salam International Center for Theoretical Physics, Strada Costiera 11, 34151 Trieste, Italy

**3** Dipartimento di Fisica "E. Pancini", Università di Napoli "Federico II", Monte S. Angelo, I-80126 Napoli, Italy

**4** Department of Physics, Lancaster University, Lancaster LA1 4YB, United Kingdom

⋆ adelmont@sissa.it

## Abstract

We propose a new theoretical method to describe the monitored dynamics of bosonic many-body systems based on the concept of the most likely trajectory. We show how such trajectory can be identified from the probability distribution of quantum trajectories, i.e. measurement readouts, and how it successfully captures the monitored dynamics beyond the average state. We prove the method to be exact in the case of Gaussian theories and then extend it to the interacting Sine-Gordon model. Although no longer exact in this framework, the method captures the dynamics through a self-consistent time-dependent harmonic approximation and reveals an entanglement phase transition in the steady state from an area-law to a logarithmic-law scaling.

## 1   Introduction

The study of phase transitions in quantum many-body systems has recently witnessed a fundamental new development: the emergence of a novel class of critical phenomena, measurement-induced phase transitions (MIPTs) [1–4]. In the simplest scenario, MIPTs arise from the interplay between unitary dynamics, which tends to build up quantum correlations within the many-body system, and quantum measurements, which, in contrast, tend to suppress them. The quest to identify and understand this new kind of critical phenomenon, while expanding rapidly in various directions, has faced fundamental experimental challenges and raised methodological and theoretical questions.

MIPTs need to be studied considering the stochastic nature of quantum measurements [5–7], and this leads to a description in terms of quantum trajectories, which represent the evolution of the quantum state conditioned on specific realizations of the measurement outcomes [8,9]. In general, MIPTs emerge only from non-linear functionals of the system's state, such as the entanglement entropy, its witnesses, or connected correlation functions. These quantities require access to beyond-average statistics of quantum trajectories and pose significant challenges from both a theoretical and an experimental points of view. Theoretical tools developed to address these issues are mainly drawn from the field of Random Matrix Theory [10–12] and disordered systems such as the replica trick [13–18]. On the experimental side, a new challenge posed by MIPTs is the full characterization of quantum states (and the estimation of complex non-linear quantities) for individual instances of an exponentially large set of quantum trajectories, a problem known as *post-selection*. While there is not a general recipe to overcome it, several studies have proposed ways to mitigate it in specific settings [19–24].

Theoretical efforts to study MIPTs have grown in several directions. A substantial body of work has focused on quantum circuits, where unitary gates are interspersed with quantum measurements, and the phase transition is typically tuned by the rate at which measurements occur in the circuit. Research in this area has explored a variety of frameworks, ranging from random circuits [2,11,25–27] to Floquet-like settings [1,28,29], leading also to several of the few experimental realizations of MIPTs [30–32]. Alongside quantum circuits, measurement-induced phase transitions have also been studied in the case of many-body Hamiltonian systems under monitoring, and this approach has allowed investigations not only into the effects of projective measurements but also into the regime of continuous weak monitoring [33,34]. Various Hamiltonian systems have been examined, including some of the most prototypical models in quantum many-body theory [24,35–38]. A significant effort has focused on fermionic systems, starting from free models [3,13,14,39–41] and moving toward more complex, interacting settings [16,42–44]. Bosonic systems have also been intensively studied [45–48], considering also connections to experimental realizations [23,24]. The nu-

merical studies conducted so far have faced the complexity of simulating stochastic dynamics for bosonic systems. The highly demanding nature of this task demands the need to approach bosonic many-body systems through analytical methods and controlled approximations. A more field-theoretical approach to bosons has been recently explored in Ref. [49], where the free boson conformal field theory (CFT) on a lattice has been studied through the lens of quantum measurements. The dynamics under weak Gaussian measurements of position and momentum has been analyzed, revealing interesting differences in quantum correlations depending on the observable measured. Notably, no measurement-induced phase transition was observed in either measurement setting.

Our paper proposes a new theoretical approach to address bosonic monitored systems. The method relies on the concept of the most likely trajectory, which enables the description of the monitoring process through a single representative trajectory. By following the dynamics of many-body systems along the most-likely trajectory, we obtain deterministic equations of motion, significantly reducing the complexity of the full stochastic problem.

We benchmark this method for Gaussian bosons, for which a closed set of stochastic equations of motion exactly describes the dynamics. Although generally difficult to solve, the set of equations can be greatly simplified using Wick's theorem. We show that our method is exact in the case of free bosons, where it reproduces the results of Ref. [49]. We use these findings as a foundation for extending the approach to the more complex framework of interacting bosons under monitoring. In this case, the stochastic nature of monitored quantum dynamics adds to the well-known challenges of treating interactions in many-body systems. This typically leads to an infinite, unclosed set of coupled stochastic equations of motion, making analytical or numerical treatment significantly harder. This deeply affects and makes untractable the system central to our study in this paper: the bosonic monitored Sine-Gordon model. On one hand, we manage the complexity arising from interactions by using a well-established approximation, the Self-Consistent Time-Dependent Harmonic Approximation (SCTDHA) [50, 51], which maps the model into a quadratic self-consistent time-dependent theory. On the other hand, we address the stochastic complexity of the model by applying our most likely trajectory method. This provides a closed set of deterministic equations, which allow for the description of a monitored many-body setting, preserving information about the measurement process, and eliminating the complexity associated with the stochastic setting, which considers all trajectories. This novel perspective enables us to investigate quantum correlations in the model, finding a measurement-induced phase transition associated with a delocalization process driven by measurements in the monitored Sine-Gordon model.

The paper is organized as follows. Section 2 introduces the setting we will use throughout the paper and derives the framework of the *most-likely trajectory* approach, building step-by-step the approximation. We then present in Section 3 how the method can exactly reproduce the results of the Free Bosons CFT as a benchmark for our method. In Sec. 4, we introduce the monitored Sine-Gordon model and the SCTDHA approximation. We carefully study its dynamical and steady state properties, particularly focusing on the behaviour of quantum correlations which signal the presence of a measurement-induced phase transition. Our conclusions about the *most-likely trajectory* approach and its prediction on the Sine-Gordon model are contained in Sec. 5.

# 2 Most-likely Trajectory Approach

This Section focuses on the derivation of the most likely trajectory approach for monitored dynamics. We will start by defining the continuous monitoring setting we want to address in this paper, which involves many independent Gaussian weak measurements. We will then

move to the introduction of two important quantities to describe quantum trajectories in this context: their conditional and joint probability distributions.

These objects are the starting points to build our method: we will analyze them carefully and identify which trajectory dominates the ensemble generated by the monitored dynamics. This leads to the construction of the most-likely trajectory and its use to analyze the monitored dynamics of bosonic many-body systems.

## 2.1 Measurements setting

Throughout the paper, we will continuously monitor bosonic systems through Gaussian weak measurements [5–7,52,53]. To keep a general description of the most likely trajectory method, we will consider measurements of an operator $\hat{R}$, with the only constraint that its eigenvectors should define a complete (or overcomplete) basis for the Hilbert space.

Measurements in the time interval $\delta t$ are implemented through the Kraus operator [5]

$$\hat{M}_r = \left(\frac{2\gamma\delta t}{\pi}\right)^{1/4} e^{-\gamma\delta t(r-\hat{R})^2}, \tag{1}$$

where $r \in \mathbb{R}$ denotes the detection's readout and $\gamma$ is the measurement strength. The operator describes the superposition of projectors over the eigenvectors of the $\hat{R}$ operator ($\hat{R}|R\rangle = R|R\rangle$), weighted by a Gaussian factor centered around $r$. The width of the distribution determines the measurement strength $\gamma$, which reproduces the projective measurement case in the limit $\gamma \to \infty$.

The repeated application of this operator induces a monitored dynamics. The evolution of an initial state $|\psi_0\rangle$ over an infinitesimal time-step $\delta t$ is obtained through the action of both measurement operator $\hat{M}_r$ and the unitary operator $\hat{\mathcal{U}}_{\delta t} = e^{-i\delta t\hat{H}}$ (setting $\hbar = 1$):

$$\left|\tilde{\psi}_{r_1}\right\rangle = \hat{\mathcal{U}}_{\delta t}\hat{M}_{r_1}|\psi_0\rangle. \tag{2}$$

The monitored evolution described in Eq. (2) is non-unitary and leads to an unnormalized state, which will be labeled with a tilde: $\left|\tilde{\psi}\right\rangle$ or $\hat{\tilde{\rho}}$. The trace of unnormalized staes corresponds to the probability of obtaining the measurement outcome $r_1$ conditioned by starting from the initial state $|\psi_0\rangle$:

$$P(r_1|\psi_0) = \text{Tr}\left\{\left|\tilde{\psi}_{r_1}\right\rangle\left\langle\tilde{\psi}_{r_1}\right|\right\}. \tag{3}$$

This allows to write the evolved physical state as $\left|\psi_{r_1}\right\rangle = \left|\tilde{\psi}_{r_1}\right\rangle/\sqrt{P(r_1|\psi_0)}$, and can be easily generalized to the case of a mixed state.

The continuous dynamics up to time $t$, is instead given by

$$\left|\tilde{\psi}_r\right\rangle = \lim_{\substack{\delta t\to 0 \\ K=\frac{t}{\delta t}\to\infty}} \prod_{k=1}^{K}\left[\hat{\mathcal{U}}_{\delta t}\hat{M}_{r_k}\right]|\psi_0\rangle. \tag{4}$$

The collection of measurement outcomes $\{r_k\}_{k=1}^{K} \xrightarrow{\delta t\to 0} r(t')$ ($t' \in [0,t]$) constitutes a quantum trajectory. Notice how now $r(t_k)$ corresponds to the measurement outcome for the application of the $k^{\text{th}}$ measurement operator. The probability of measuring the trajectory $r$ - i.e. the joint probability of the collection of measurement readouts up to time $t$ given the initial state - is obtained by tracing the unnormalized fully evolved state.

$$P[r|\psi_0;t] = \text{Tr}\left\{\left|\tilde{\psi}_r\right\rangle\left\langle\tilde{\psi}_r\right|\right\}. \tag{5}$$

137  This can be understood in terms of single steps' conditional probabilities, Eq. (3). For two time
138  steps, we have

$$
\begin{aligned}
P[\{r_1, r_2\} | \psi_0; t_2] &= P(r_2|r_1, \psi_0) P(r_1|\psi_0) \\
&= \mathrm{Tr}\{\hat{M}_{r_2} |\psi_{r_1}\rangle\langle\psi_{r_1}| \hat{M}_{r_2}\} \mathrm{Tr}\{|\tilde{\psi}_{r_1}\rangle\langle\tilde{\psi}_{r_1}|\} \\
&= \mathrm{Tr}\{\hat{M}_{r_2} \hat{\mathcal{U}}_{\delta t} \hat{M}_{r_1} |\psi_0\rangle\langle\psi_0| \hat{M}_{r_1} \hat{\mathcal{U}}_{\delta t}^\dagger \hat{M}_{r_2}\}.
\end{aligned}
\tag{6}
$$

139  Repeating this for $K = t/\delta t$ steps reproduces Eq. (5).

140      The conceptual steps made above allow one to define the conditional and joint probability
141  distributions for quantum trajectories.

142      The **conditional probability distribution** for obtaining the outcome $r$ given that the sys-
143  tem is in the state $|\psi\rangle$, can be written in $R-$representation as

$$
P(r|\psi) = \left(\frac{2\gamma\delta t}{\pi}\right)^{1/2} \int dR \, e^{-2\gamma\delta t (r-R)^2} |\psi(R)|^2,
\tag{7}
$$

144  where $\psi(R) = \langle R|\psi\rangle$.

145      Having continuous monitoring allows for the expression of the **Joint probability distribu-
146  tion** of a string of readouts up to time $t$ in terms of a Path Integral Formalism [54]. In order
147  to derive it, we consider both the operator $\hat{R}$ and its canonically conjugated operator. Simple
148  examples are position measurements $\hat{R} = \hat{x}$, with the canonically conjugated $\hat{p}$ momentum as
149  in Ref. [54], or momentum measurements $\hat{R} = \hat{p}$ with the position $\hat{x}$ being the conjugate as we
150  will consider throughout this paper. Then, in the $R-$representation, starting from the initial
151  state $\hat{\rho}_0$ and taking the continuous limit $\delta t \to 0$ in equations (4) and (5), we get:

$$
P[r; t] = \int dR_0 \, dR_0' \, dR \, \rho(R_0, R_0') \int_{R_1(0)=R_0}^{R_1(t)=R} \mathcal{D}R_1(t') \int_{R_2(0)=R_0'}^{R_2(t)=R} \mathcal{D}R_2(t') \, e^{iS_0[R_1;t] - iS_0[R_2;t] + S_{\mathrm{meas}}[R_1,R_2;t]}
$$

$$
S_{\mathrm{meas}}[R_1, R_2; t] = -\gamma \int_0^t dt' \left[ (r(t') - R_1(t'))^2 + (r(t') - R_2(t'))^2 \right]
\tag{8}
$$

152      Where $S_0[R_1; t]$ and $S_0[R_2; t]$ are the actions (in $R-$representation) given by unitary dy-
153  namics along the forward and backward Keldysh branches. The above expression simply repre-
154  sents the evolution of the un-normalized state along the trajectory $r(t')$. The effect of the mea-
155  surement in this framework corresponds to a coupling of the forward and backward branches
156  through the imposition of the same trajectory $r(t')$ along the two branches. Details on the
157  derivation of the path integral expression can be found in Appendix A. Despite the joint prob-
158  ability distribution explicitly depending on the initial state $\hat{\rho}_0$, we omit it in the expression
159  $P[r; t]$ to simplify the notation.

160      Now, according to Born's rule, we can compute averages of any trajectory functional $F[r]$
161  over this probability distribution as

$$
\overline{F[r]} = \int \mathcal{D}r(t') P[r; t] F[r],
\tag{9}
$$

162  where the measure of the path integration over trajectories is defined as

$$
\int \mathcal{D}r(t') = \lim_{\substack{\delta t \to 0 \\ K \to \infty}} \prod_{k=1}^{K} \left(\frac{2\gamma\delta t}{\pi}\right)^{1/2} \int_{-\infty}^{+\infty} dr_k.
\tag{10}
$$

163      The ultimate goal of the study of monitored systems is being able to identify measurement-
164  induced phase transitions. The established diagnostics requires the use of the trajectory aver-
165  age of non-linear objects in the system state $\hat{\rho}_r$. Typical examples are averages of variances

$F[r] = \text{Tr}\{\hat{O}^2 \hat{\rho}_r\} - \text{Tr}\{\hat{O}\hat{\rho}_r\}^2$ or entropies such as $F[r] = -\text{Tr}\{\hat{\rho}_r \log \hat{\rho}_r\}$. These objects generally have a very complex structure in $r(t')$, and the first complication that one usually finds in this kind of computation lies in the path integration over the trajectories.

The tool, which is typically introduced in this context to deal with the complications generated by the path integral over trajectories is the Replica Trick [13–18]. It allows simplifications of Eq. (9) at the price of considering the dynamics of many copies of the state $\hat{\rho}_r$ sharing the same sequence of measurement outcomes.

In the following part of the Section, we will introduce a different method to reduce the complexity generated by dealing with the full ensemble of trajectories.

## 2.2 The most likely trajectory

Our idea relies on the study of the structure of the probability distribution we have introduced in the previous Section. Specifically, we aim to simplify the complexity of computing averages over the whole probability distribution by determining which trajectory dominates the distribution and characterizing the monitored dynamics in terms of that trajectory only.

We first build the most likely trajectory starting from a single time step perspective and restoring again the more general setting which measures $\hat{R}$. Suppose the system is in the state $|\psi\rangle$, the conditional probability distribution for Gaussian momentum measurements (7) can be approximated as

$$P(r) \sim \left(\frac{2\gamma\delta t}{\pi}\right)^{1/2} e^{-2\gamma\delta t(r-\langle\hat{R}\rangle_\psi)^2}. \tag{11}$$

The conceptual step in this approximation lies in noticing that in eq. (7) the Gaussian $e^{-2\gamma\delta t(r-R)^2}$ will be much broader than the wavefunction's probability distribution for $\delta t \to 0$, such that [5]

$$|\psi(R)|^2 \sim \delta(R - \langle\hat{R}\rangle_\psi), \tag{12}$$

which holds for unimodal wavefunctions.

From the previous approximation, we notice that the most likely measurement readout for the time step corresponds to $r^* = \langle\hat{R}\rangle_\psi$. For Gaussian measurements this also corresponds to the average value $\overline{r} = \int dr\, r\, P(r)$.

It is now possible to build the full trajectory out of single time steps:

$$r^*(t) \quad = \quad \overline{r(t)} = \langle\hat{R}\rangle_{\psi^*(t)} \tag{13}$$

$$\text{with } |\psi^*(t)\rangle \quad \propto \quad \lim_{\substack{\delta t \to 0 \\ K = t/\delta t \to \infty}} \prod_{k=1}^{K-1} \left[\hat{\mathcal{U}}_{\delta t} \hat{M}_{r_k^*}\right] |\psi_0\rangle. \tag{14}$$

This means that the most likely trajectory is built using the step-by-step procedure:

- Select $r_k^*$ as the most likely value of the conditional probability distribution for $|\psi^*(t'-\delta t)\rangle$;

- Impose the evolution of the state to be conditioned on the $r_k^*$ value and update the state: $|\psi^*(t')\rangle \propto \hat{\mathcal{U}}_{\delta t} \hat{M}_{r_k^*} |\psi^*(t'-\delta t)\rangle$

- Obtain the next step by repeating the whole procedure, starting with the evolved state $|\psi^*(t')\rangle$ as new initial state;

Notice that the conditional probability distribution for $r$ in Eq. (7) can be recast in the classic form of Wiener stochastic processes, as the the measurement outcome can be expressed as [5,6]

$$r = \langle\hat{R}\rangle_\psi + \frac{dW}{2\sqrt{\gamma}\delta t}, \quad \overline{dW} = 0, \quad dW^2 = \delta t, \tag{15}$$

In this framework, following the most likely trajectory means imposing

$$dW = dW^* = 0 \quad \forall t. \tag{16}$$

Notice that this implies $dW^2 = 0$. This differs from the standard Ito rule $dW^2 = \delta t$, which produces additional deterministic terms in the master equation.

Once the most likely trajectory is identified, we are able to study the evolution of the system conditioned to that trajectory only. The master equation to evolve the quantum state can be obtained from a linear expansion in $\delta t$ of a one-step evolution equation

$$\rho_{\mathbf{r}}^*(t + \delta t) = \frac{\hat{\mathcal{U}}_{\delta t} \hat{\mathcal{M}}_{\mathbf{r}^*} \hat{\rho}_{\mathbf{r}^*}(t) \hat{\mathcal{M}}_{\mathbf{r}^*} \hat{\mathcal{U}}_{\delta t}^\dagger}{\text{Tr}\{\hat{\mathcal{M}}_{\mathbf{r}^*} \hat{\rho}_{\mathbf{r}^*}(t) \hat{\mathcal{M}}_{\mathbf{r}^*}\}}, \tag{17}$$

which yields for the measurement setting we have introduced:

$$\begin{aligned}
\dot{\rho}^*(t) &= -i\left[\hat{H}, \rho^*(t)\right] - \gamma \sum_{j=1}^{M} \left\{\hat{R}_j^2 - \left\langle\hat{R}_j^2\right\rangle_{\rho^*(t)}, \rho^*(t)\right\} \\
&\quad + 2\gamma \sum_{j=1}^{M} \left\langle\hat{R}_j\right\rangle_{\rho^*(t)} \left\{\hat{R}_j - \left\langle\hat{R}_j\right\rangle_{\rho^*(t)}, \rho^*(t)\right\}.
\end{aligned} \tag{18}$$

where we have relabeled $\hat{\rho}_{\mathbf{r}^*}$ as $\hat{\rho}^*$ for simplicity.

Notice that the master equation we have obtained is trace preserving by construction, deterministic, and is non-linear in the system's state. These last aspects are what fundamentally makes the most likely trajectory (average trajectory $dW = 0$) approach different from the Lindblad (average state $\bar{\hat{\rho}}$) master equation and the state-diffusion master equation ($\overline{dW} = 0$, $dW^2 = \delta t$), which instead considers the full statistics of trajectories and loses information about the specific measurement procedure.

This equation fully captures the power of this approximation: the master equation is now deterministic and preserves information about the measurement process.

Considering the whole evolution, the procedure we have presented is also understood in terms of a Saddle Point of the joint probability distribution setting. We consider a more general setting, and suppose now to monitor *independently* a collection of operators $\hat{R}_i$ $i = 1, .., M$, i.e. our measurement operator becomes

$$\hat{\mathcal{M}}_{\mathbf{r}} = \prod_{i=1}^{M} \hat{M}_{r_i}, \tag{19}$$

where $\mathbf{r} = (r_1, ..., r_M)$ contains the measurement outcomes associated to the independent measurements of the set of $\{\hat{R}_i\}$. Notice that in this context $i$ labels the $i^{\text{th}}$ operator and does not label time anymore. The generalization of the joint probability distribution in Eq. (8) for $M$ independent measurements is trivial and can be found in App. A.

From the conditional probability setting, we have noticed how the most-likely measurement outcome also corresponds to the average one, in this case $\overline{\mathbf{r}(t)} = \mathbf{r}^*(t) = \left\langle\hat{\mathbf{R}}\right\rangle_{\psi_r^*}$. Thus we start from a simple case and calculate the average trajectory according to the joint probability distribution

$$\overline{\mathbf{r}(t)} = \int \mathcal{D}\mathbf{r}(t') \mathbf{r}(t) P[\mathbf{r}; t], \tag{20}$$

where $t$ is the final time of the path integral.

We introduce the fictitious action

$$\mathcal{S}[\mathbf{r}; t] = \log P[\mathbf{r}; t], \tag{21}$$

230 and we proceed via Saddle Point approximation for the integration over $r(t)$.

$$\overline{r_j(t)} \sim r_j^*(t), \quad \text{with} \ r_j^*(\tau): \left. \frac{\partial \mathcal{S}[\mathbf{r};t]}{\partial r(\tau)} \right|_{r_j^*(\tau)} = 0 \tag{22}$$

231 As calculated explicitly in App. A one finds:

$$\mathbf{r}^*(t) = \left\langle \hat{\mathbf{R}} \right\rangle_{\hat{\rho}_{\mathbf{r}^*(t)}}. \tag{23}$$

232 Notice that this result corresponds exactly to what we have found from the conditional prob-
233 ability distribution in Eq. (13). When the fictitious action $\mathcal{S}[r]$ is quadratic, usually for free
234 theories subject to Gaussian measurements, the Saddle Point method is exact. In these cases,
235 the most likely trajectory (coinciding with the average in this case) is a good representative
236 trajectory of the full ensemble as it dominates it [53, 55].
237     Additionally, notice how the saddle point approximation we have performed highly de-
238 pends on the function we are averaging over. For a general non-linear functional of the quan-
239 tum trajectory $F[r]$, the average $\overline{F[r]} = \int \mathcal{D}r(t') F[r] P[r]$ might produce a different Saddle
240 Point trajectory. However, as long as the structure of $F[r]P[r]$ remains the same of $P[r]$ alone,
241 which is usually the case for Gaussian theories as we will see in the next Section, the Saddle
242 point will remain the one we found.
243     This formalism allows however to go beyond the Saddle Point and consider Gaussian fluctu-
244 ations around the Saddle Point solution. We first notice that the Gaussian fluctuations around
245 the Saddle point solutions approximate the action up to second order as

$$\mathcal{S}[r] \sim \mathcal{S}[\mathbf{r}^*] + \frac{1}{2} \int dt' dt'' (\mathbf{r}(t') - \mathbf{r}^*(t')) \mathbb{S}^{(2)}[\mathbf{r}^*; t, t'] (\mathbf{r}(t'') - \mathbf{r}^*(t'')), \tag{24}$$

246 where we have introduced

$$\mathbb{S}_{i,j}^{(2)}[\mathbf{r}^*; t', t''] = \left. \frac{\partial^2 \mathcal{S}[r(t)]}{\partial r_j(t') \partial r_i(t'')} \right|_{\mathbf{r}^*} = -4\gamma \delta_{i,j} \delta(t - t') + 16\gamma^2 \sigma_{R_i,R_j}^{++}(t', t''). \tag{25}$$

247 Above we have defined $\sigma_{R_i R_j}^{++}(t, t') = \left\langle \hat{R}_j^+(t) \hat{R}_i^+(t') \right\rangle_{\rho^*} - \left\langle \hat{R}_j^+(t) \right\rangle_{\rho^*} \left\langle \hat{R}_i^+(t') \right\rangle_{\rho^*}$ with $\hat{R}^+$ being
248 the classical Keldysh component of the field as detailed in App. A. This allows us to estimate
249 corrections to the Saddle Point solution for the average trajectory as

$$\overline{r(t)} \sim r^*(t) + \Delta r = r^*(t) + \sqrt{\overline{(r(t) - r^*(t))^2}}. \tag{26}$$

250 In general the fluctuations are thus determined by the inverse of $\mathbb{S}^{(2)}(r^*; t, t')$, which is a
251 rather complicated object consisting of both a local term and a term involving two-point
252 correlations evaluated along the most likely trajectory itself. However, in the strong mea-
253 surements limit, we can consider $\sigma_{R_i, R_j}^{++} \sim 0$ as projective measurements tend to decorrelate
254 the field we are measuring, both in time and space. Thus, a good estimate for the fluctua-
255 tions in the strong measurement limit is only obtained from to the local part of $\mathbb{S}^{(2)}$, yielding
256 $\mathbb{S}^{sm}(r^*; t, t') \sim -4\gamma \delta_{ij} \delta(t - t')$, and a correction to the saddle point solution of the form

$$\overline{r(t)} \sim r^*(t) + \Delta r = r^*(t) + \frac{\mathcal{A}}{\gamma}, \tag{27}$$

257 where $\mathcal{A}$ amounts to a UV regularization in momentum space. Our approach becomes increas-
258 ingly accurate in the strong-measurements limit.

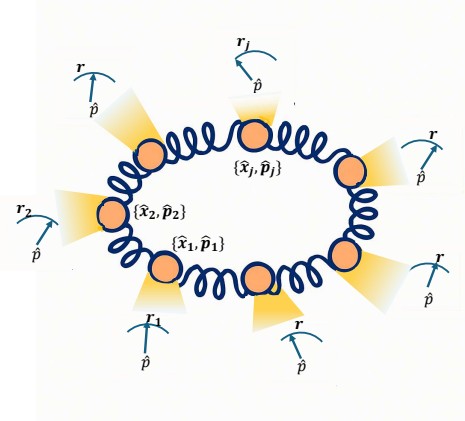

Figure 1: Representation for the Free Bosons CFT for the specific case of $N = 7$. The picture shows a chain of harmonic oscillators with periodic boundary conditions. Each oscillator, represented with a red sphere, has its momentum $\hat{p}_i$ measured independently at each time step from distinct measurement devices.

## 3   Benchmark: Monitored Free Bosons CFT

We will start by benchmarking our method against a free theory. We consider a Free Bosons model, already studied in Ref. [49]. In his lattice version, the Hamiltonian for $N$ sites reads:

$$\hat{H}_{\text{FB}} = \frac{\omega}{2} \sum_{j=1}^{N} \left( \hat{p}_j^2 + (\hat{x}_{j+1} - \hat{x}_j)^2 + r_N^2 \, \hat{x}_j^2 \right), \tag{28}$$

with $r_N^2 = \Omega/N/\omega$ being a cutoff to normalize the spectrum at zero momentum and $\left[ \hat{x}_j, \hat{p}_{j'} \right] = i \delta_{j,j'}$. We consider periodic boundary conditions. The Hamiltonian essentially describes a massless model for free bosons and is represented in Fig. 1.

We consider weak momentum measurements occurring independently at each site by setting $\hat{R}_i = \hat{p}_i$ for $i = 1, .., N$ from this Section on. The measurement operator is given by

$$\hat{\mathcal{M}}_{\mathbf{r}} = \prod_{i=1}^{N} \hat{M}_{r_i} = \prod_{i=1}^{N} \left( \frac{2\gamma \delta t}{\pi} \right)^{1/4} e^{-\gamma \delta t (r_i - \hat{p}_i)^2}. \tag{29}$$

The monitored dynamics is described through the most-likely trajectory approach through the simple master equation

$$
\begin{aligned}
\dot{\rho}^*(t) = & -i \left[ \hat{H}_{\text{FB}}, \rho^*(t) \right] - \gamma \sum_{j=1}^{N} \left\{ \hat{p}_j^2 - \left\langle \hat{p}_j^2 \right\rangle_{\rho^*(t)}, \hat{\rho}^*(t) \right\} \\
& + 2\gamma \sum_{j=1}^{N} \left\langle \hat{p}_j \right\rangle_{\rho^*(t)} \left\{ \hat{p}_j - \left\langle \hat{p}_j \right\rangle_{\rho^*(t)}, \hat{\rho}^*(t) \right\}.
\end{aligned} \tag{30}
$$

Exploiting the Gaussianity of both the measurements and unitary dynamics, we can choose as initial state $|\psi_0\rangle$ a many-body Gaussian state, and describe its evolution through the averages $\langle \hat{x}_i \rangle$, $\langle \hat{p}_i \rangle$ and connected correlations $\sigma_{xp}^{i,j} = \frac{1}{2} \left\langle \{ \hat{x}_i, \hat{p}_j \} \right\rangle - \langle \hat{x}_i \rangle \langle \hat{p}_j \rangle$, $\sigma_{xx}^{i,j}$, $\sigma_{pp}^{i,j}$ only, as Gaussianity is preserved by this kind of dynamics. Notice moreover that the specifics of the initial state do not matter in our case, as we will be interested in the steady-state features only.

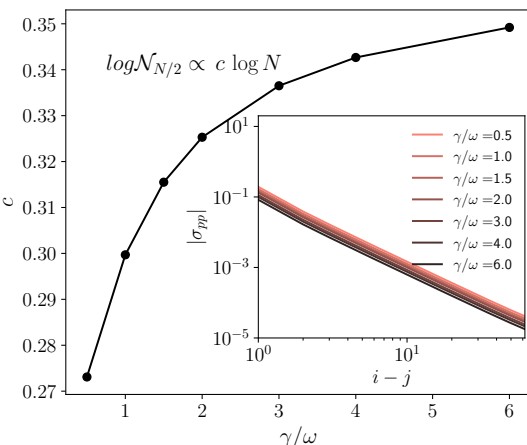

Figure 2: Logarithmic Negativity for the Free Bosons CFT lattice model, the plot shows the coefficient for the fit $\log \mathcal{N}_{N/2} \sim c \log N$, which is always non-zero in this case. Inset: power-law decaying correlations in log-log scale.

Ref. [49] predicts for this specific setting a trajectory-averaged power-law decaying $\sigma_{pp}^{i,j}$ correlations along with a bipartite logarithmic negativity scaling as $\log \mathcal{N} \sim \log N$ for all measurement strengths $\gamma$, and hence the absence of a measurement-induced phase transition. The exact stochastic and trajectory-averaged equations to evolve averages and connected correlations can be found in Appendix B. An important feature to keep in mind is that this exact approach yields, at the level of trajectories, stochastic equations for the averages $\langle \hat{x}_j \rangle, \langle \hat{p}_j \rangle$ and deterministic equations for the correlations. This characteristic is peculiar to free bosonic systems. We can now try to reproduce the trajectory-averaged results with our method. The master equation to evolve the many-body state along the most-likely trajectory is obtained by in inserting the Free Bosons Hamiltonian (28) into the master equation derived in Eq. (30).

Exploiting the Gaussianity through Wick theorem, we can derive the deterministic Heisenberg equations to determine the evolution of averages and correlations:

$$\partial_t \langle \hat{x}_i \rangle = \omega \langle \hat{p}_i \rangle \tag{31a}$$

$$\partial_t \langle \hat{p}_i \rangle = -2\omega \langle \hat{x}_i \rangle + \omega (\langle \hat{x}_{i+1} \rangle + \langle \hat{x}_{i-1} \rangle) \tag{31b}$$

$$\partial_t \sigma_{xx}^{i,j} = \omega \left( \sigma_{xp} + \sigma_{px} \right)^{ij} - 4\gamma (\sigma_{xp} \sigma_{xp}^T)^{ij} + \gamma \delta_{ij} \tag{31c}$$

$$\partial_t \sigma_{pp}^{i,j} = \omega \left[ -(1 + \frac{r_N^2}{2}) \sigma_{xp}^{i,j} + (\sigma_{xp}^{i-1,j} + \sigma_{xp}^{i+1,j}) + (i \leftrightarrow j) \right] - 4\gamma (\sigma_{pp}^2)^{ij} \tag{31d}$$

$$\partial_t \sigma_{xp}^{i,j} = \omega (\sigma_{pp}^{i,j} - (2 + r_N^2) \sigma_{xx}^{i,j} + \sigma_{xx}^{i,j-1} + \sigma_{xx}^{i,j+1}) - 2\gamma ((\sigma_{xp} + \sigma_{px}) \sigma_{pp})^{ij} \tag{31e}$$

As can be checked by comparing these equations with Appendix B reporting the exact results, the evolution along the most-likely trajectory for a free bosonic system is the same as the exact trajectory-averaged equations. In particular, Eq.s (31a), (31b) reproduce exactly the trajectory average of the stochastic equations found by [49], and Eq.s (31c)- (31e) reproduce exactly the deterministic equations found in the same Reference. This fact is strikingly important: the most likely trajectory fully encodes the properties of the trajectory-averaged monitored dynamics of free bosonic systems for what concerns variances. Eq.s (B.2c)- (B.2e) contain information about the full stochastic process (characterized by the average $\overline{dW} = 0$ and fluctuations $dW^2$), while our equations contain information about the average/most-likely value only. Yet the equations for the variances are the same and produce exactly the same steady-state values.

Trivially, our equations also reproduce the exact steady-state predictions of Ref. [49], proving again the absence of a measurement-induced phase transition. To show this, we study the scaling of $\sigma_{pp}$ correlations and the logarithmic negativity of the steady state of the system in Fig. 2. The latter object quantifies the quantum correlations between bipartitions of the system and is defined as

$$\log \mathcal{N}_{N/2} = \log \left\{ \mathrm{Tr} \left\{ \hat{\rho}_{\mathrm{SS}}^{T_{N/2}} \right\} \right\}, \tag{32}$$

where $\hat{\rho}_{\mathrm{SS}}^{T_{N/2}}$ is the partial transpose of a bipartition of the many-body density matrix in the steady-state. The peculiarity of a Gaussian system is that the logarithmic negativity can be expressed directly in terms of $\hat{x}_j$ and $\hat{p}_j$ correlations [56–58], as reported in Appendix B.

These calculations show that our method is exact for free bosonic theories. Having a quadratic theory, makes the joint probability distribution $P[\mathbf{r}; t]$ quadratic in $\mathbf{r}$ in turn. This means that the Saddle Point over the fictitious trajectory action $\mathcal{S}[\mathbf{r}; t]$ is exact, allowing the correct identification of the dominant trajectory in the ensemble.

It should also be noticed that bosonic Gaussian theories can rely on another useful property. The calculation we have performed in this section essentially can be summed up in

$$\overline{F_k[\mathbf{r}(t)]} = \int \mathcal{D} \mathbf{r}(t') F_k[\mathbf{r}(t)] P[\mathbf{r}; t] \sim F_k[\mathbf{r}^*(t)],$$

with $a = 1, 2$ and $F_1[\mathbf{r}] = \left\langle \hat{R} \right\rangle_{\hat{\rho}^*}$, $F_2[\mathbf{r}] = \left\langle \hat{R} \right\rangle_{\hat{\rho}^*}^2$. For free theories the structures in $\mathbf{r}$ generated by the products $F_1[r(t)]P[\mathbf{r}; t]$ and $F_2[r(t)]P[\mathbf{r}; t]$ is always the same as in Eq. 25. Thus, the (exact) saddle Point which determines the most probable trajectory is also a saddle point for $F_1[r(t)]P[\mathbf{r}; t]$ and $F_2[r(t)]P[\mathbf{r}; t]$.

In the remaining part of the paper, we will test and use our method for an interacting system, where the approach is no longer exact. We will characterize it in the bosonic Sine-Gordon model framework, highlighting the differences from the non-interacting case and getting some novel results within our most likely trajectory approach.

# 4 Monitored interating bosons: The Sine-Gordon Model

## 4.1 The model

To explore the applicability of the most likely trajectory approach, we study the monitored dynamics of an interacting model content by adding a Sine-Gordon potential [59, 60] to the free Gaussian model in Sec. 3. Specifically

$$\hat{H} = \hat{H}_{\mathrm{FB}} + \hat{H}_{\mathrm{int}},$$
$$\hat{H}_{\mathrm{int}} = -\frac{J}{\alpha^2} \sum_{j=1}^{N} \cos\left(\alpha \hat{x}_j\right), \tag{33}$$

with $\hat{H}_{\mathrm{FB}}$ given in Eq. (28). We monitor the model through weak measurements of the momentum operator at each site, described again by the operator from Eq. (29). A representation of the model is presented in Fig. 3(a).

A few comments are in order before analyzing the properties of the model.

- In the $\alpha \to \infty$ we obtain again the Free Bosons CFT model of Ref. [49]. In this limit, we know the most-likely trajectory approach to be exact as the model is free.

- For finite $\alpha$, the interaction Hamiltonian tends to localize the $j^{\mathrm{th}}$ oscillator in one of the wells of the cosine potential. Momentum $\hat{p}_j$ measurements tend to delocalize in

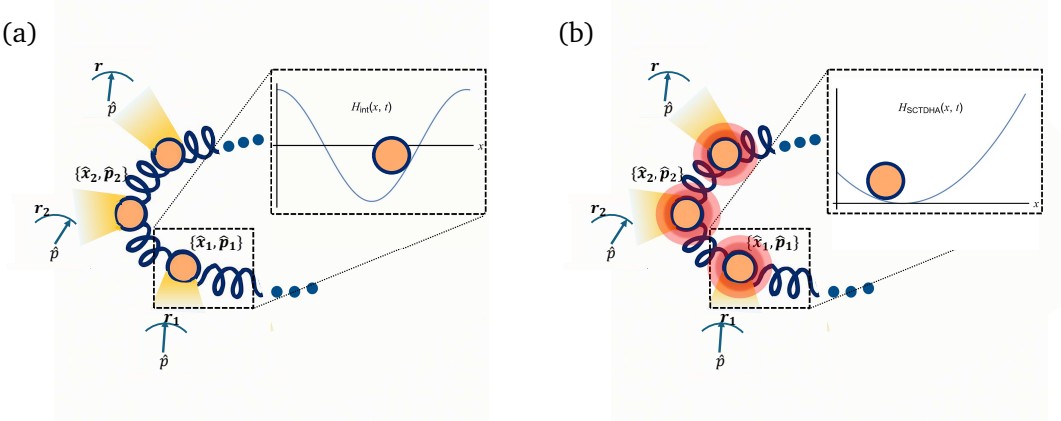

Figure 3: (a) Truncated representation for the Sine-Gordon model for the specific case of $N = 7$. The picture shows a chain of harmonic oscillators with periodic boundary conditions. Each oscillator, represented with a red sphere, is subject independently to a cosine potential as shown for the zoomed window and has its momentum $\hat{p}_i$ measured at each time step from distinct measurement devices. (b) Truncated representation for the Sine-Gordon model in the SCTDHA for $N = 7$. The picture shows a chain of 'dressed' harmonic oscillators in periodic boundary conditions. Each oscillator, represented with a red sphere, has its own time-dependent effective mass (the shade in the picture) and has its momentum $\hat{p}_i$ measured at each time step from distinct measurement devices. The zoomed picture shows the modified potential for a fixed time.

contrast, as shown in the Free Bosons model. We expect these two effects to compete and eventually lead to a measurement-induced phase transition.

The master equation generating the system's evolution can be directly obtained from Eq. (30) adding the interaction Hamiltonian from Eq. (33). As trivial for an interacting theory, the Heisenberg equations for $\hat{x}_i$ and $\hat{p}_i$ include all the moments of each operator due to the commutator with $\hat{H}_{\text{int}}$. The equations cannot be closed anymore at the level of second-order cumulants as in the case of a free theory.

This issue is unrelated to the monitoring and, among the various approximations developed to address it, we use the Self-Consistent-Time-Dependent Harmonic Approximation (SCT-DHA). The SCTDHA is particularly suited to our case since it self-consistently reduces the theory to a Gaussian one, where monitoring can be accounted for exactly as in Sec. 3. Specifically, the SCTDHA approximates self-consistently the evolution generated by the interaction term with a quadratic time-dependent one, i.e.

$$\hat{\mathcal{U}}_t = e^{-i(\hat{H}_{\text{FB}} + \hat{H}_{\text{int}})t} \sim e^{-i\hat{H}_{\text{FB}} t - i \int_0^t dt' \hat{\tilde{H}}_{\text{int}}(t')}. \tag{34}$$

The method is commonly used in the framework of the Sine-Gordon model and has been tested both at equilibrium [50] and out-of-equilibrium in the setting of quantum quenches [51].

The time-dependent approximation of the interaction Hamiltonian is

$$
\hat{\tilde{H}}_{\text{int}}(t) = \sum_{j=1}^{N} \Big( f_j(t) + g_j(t)\hat{x}_j + h_j(t)\hat{x}_j^2 \Big)
$$
$$
g_j(t) = \frac{J}{\alpha} e^{-\frac{\alpha^2}{2}\sigma_{xx}^{j,j}(t)} \Big( \sin\!\big(\alpha\langle\hat{x}_j\rangle_t\big) - \alpha\langle\hat{x}_j\rangle_t \cos\!\big(\alpha\langle\hat{x}_j\rangle_t\big) \Big) \tag{35}
$$
$$
h_j(t) = \frac{J}{2} e^{-\frac{\alpha^2}{2}\sigma_{xx}^{j,j}(t)} \cos\!\big(\alpha\langle\hat{x}_j\rangle_t\big)
$$

where $\langle\bullet\rangle_t$ and $\sigma_{xx}^{jj}(t)$ are the average and variance of the $\hat{x}_j$ operator calculated over the state at time $t$. The term $\sum_j f_j(t)$ doesn't affect the dynamics of the system, and for this reason, it does not need to be specified.

A detailed calculation about the derivation of the approximated interaction Hamiltonian for our specific model can be found in Appendix C. We report in this section the conceptual steps used at each time step to derive the SCTDHA:

1. A shift of the operators:

$$
\hat{\xi}_j = \hat{x}_j - \langle\hat{x}_j\rangle_t, \ \text{with} \ \langle\hat{\xi}_j\rangle_t = 0.
$$

2. A second-order expansion of the interaction Hamiltonian:

$$
\tilde{H}'_{\text{int}} \sim \sum_{j=1}^{N} \Big( \mathcal{C}_j(t) + \hat{\mathcal{V}}_j(t)\hat{\xi}_j + \hat{\mathcal{M}}_j(t)\hat{\xi}_j^2 \Big)
$$

3. A Hartree factorization $\hat{\mathcal{V}}_j \to \langle\hat{\mathcal{V}}_j\rangle_t$, $\hat{\mathcal{M}}_j \to \langle\hat{\mathcal{M}}_j\rangle_t$.

The expression of the coefficients $\langle\hat{\mathcal{V}}_j\rangle$ and $\langle\hat{\mathcal{M}}_j\rangle$, along with the procedure to recast them into Eq. (35) can be found in App. C.

An important consequence of this approximation is that it incorporates the interaction effects in the time dependence of the parameters $f_j, g_j, h_j$ while making, at each time step, the Hamiltonian fully quadratic. We are now describing a quadratic model of bosons having an effective time-dependent mass $h_j(t)$ and a time-dependent drive $g_j(t)$. We can visualize it as in Fig. 4(b). This quadratic effective description of the Sine Gordon model enables the closure of the Heisenberg equations at the level of variances of $\hat{x}_j$ and $\hat{p}_j$ operators, as we can use Wick theorem to decouple higher moments. We stress that this approximation goes beyond a simple quadratic expansion of the interaction Hamiltonian $\hat{H}_{\text{int}} \sim J/\alpha^2 - J\hat{x}_j^2/2$ as, at each time step, the SCTDHA finds the best quadratic approximation of the exact Hamiltonian, taking into account the correlation structure of the state at each time.

Before starting the analysis of the dynamical and steady-state properties of the Sine-Gordon model through the lens of the most-likely trajectory approach, we analyze the equation of motions coming from quantum state diffusion to describe the model in the SCTDHA. This will immediately highlight the power of the most-likely trajectory approach in solving interacting systems.

374    From quantum state diffusion dynamics, we obtain:

$$d \langle \hat{x}_i \rangle (dW) = dt\, \omega \langle \hat{p}_i \rangle + 2\sqrt{\gamma} \sum_{j=1}^{N} dW_j \, \sigma_{xp}^{ji}, \tag{36a}$$

$$d \langle \hat{p}_i \rangle (dW) = dt \left[ -g_i(t; dW) - 2\Big( \omega + h_i(t; dW) \Big) \langle \hat{x}_i \rangle + \omega \Big( \langle \hat{x}_{i+1} \rangle + \langle \hat{x}_{i-1} \rangle \Big) \right] +$$

$$+ 2\sqrt{\gamma} \sum_{j=1}^{N} dW_j \, \sigma_{pp}^{ji}, \tag{36b}$$

$$\partial_t \sigma_{xx}^{i,j}(dW) = \omega \left( \sigma_{xp} + \sigma_{px} \right)^{ij} - 4\gamma (\sigma_{xp}\sigma_{xp}^T)^{ij} + \gamma \delta_{ij}, \tag{36c}$$

$$\partial_t \sigma_{pp}^{i,j}(dW) = -\omega(2 + r_N^2)\sigma_{xp}^{i,j} + \left[ \omega(\sigma_{xp}^{i-1,j} + \sigma_{xp}^{i+1,j}) - 2\sigma_{xp}^{i,j} h_j(t; dW) + (i \leftrightarrow j) \right] +$$

$$- 4\gamma(\sigma_{pp}^2)^{ij}, \tag{36d}$$

$$\partial_t \sigma_{xp}^{i,j}(dW) = \omega(\sigma_{pp}^{i,j} - (2 + r_N^2)\sigma_{xx}^{i,j} + \sigma_{xx}^{i,j-1} + \sigma_{xx}^{i,j+1}) - 2h_j(t; dW)\sigma_{xx}^{i,j} +$$

$$- 2\gamma((\sigma_{xp} + \sigma_{px})\sigma_{pp})^{ij}. \tag{36e}$$

375    The key point lies in the stochasticity of $\langle \hat{x}_i \rangle$. The parameters $h_i$ and $g_i$ of the SCTDHA
376    depend crucially on $\langle \hat{x}_i \rangle$ itself, which makes the effective mass and the external drive stochastic
377    in turn. Since the correlations $\boldsymbol{\sigma}_{xx}, \boldsymbol{\sigma}_{xp}, \boldsymbol{\sigma}_{pp}$ depend again on the parameters of the SCTDHA,
378    we are obtaining stochastic correlations.

379    This is a significant aspect of the model we are presenting. We are working with an inter-
380    acting model through the SCTDHA lenses: while the Hamiltonian $\hat{H}_{\text{int}}$ is formally Gaussian,
381    the self-consistent equations determining the parameters of the quadratic version of the inter-
382    action keep information about the interaction, this manifests in the monitored dynamics by
383    yielding stochastic equations along trajectories for the correlations, as characteristic of inter-
384    acting systems, and in sharp contrast to the free bosons CFT model.

385    Having stochastic correlations is thus a result of both the complexities of the monitored
386    problem and the presence of interactions. The most important consequence of this fact is that
387    now the set of equations (36), despite being close, cannot be solved analytically and requires
388    a numerical simulation of the whole body of trajectories.

389    The complexity of the problem is significantly reduced by the most-likely trajectory ap-
390    proach, as we will see in the next paragraphs.

## 4.2   Dynamics

392    We will now use the master equation of the SCTDHA in the most-likely trajectory setting,
393    adding the transformed interaction Hamiltonian $\hat{H}_{\text{int}} \to \tilde{H}_{\text{int}}$ to Eq. (30).

394    Before proceeding, note the differences between the Free Bosons CFT case and the Sine-
395    Gordon model in the SCTDHA. Despite being both quadratic theories, the Sine Gordon model
396    presents additional time-dependent terms in the Hamiltonian: the $j^{\text{th}}$ oscillator experiences a
397    drive $g_j(t)$ and an effective mass given by $h_j(t)$. Both terms break the scale invariance, which
398    characterizes the free boson CFT model, and this hints at a possible breaking of the logarithmic
399    Negativity scaling as $\log N$ discussed in [49]. Notice that, coherently with what observed for
400    the exact sine Gordon model (33), the limit $\alpha \to \infty$ in Eq. (35) suppresses the effective mass
401    and drive, resulting again in the Free Boson CFT model.

402    From the master equation, we can derive the equations of motion for along the most-
403    likely trajectory for $\langle \hat{x}_i \rangle_t$, $\langle \hat{p}_i \rangle_t$, $\sigma_{xp}^{i,j}(t)$, $\sigma_{xx}^{i,j}(t)$, $\sigma_{pp}^{i,j}(t)$, which in the context of the SCTDHA
404    again fully characterize the system's many-body state. From now on, we will omit the time-

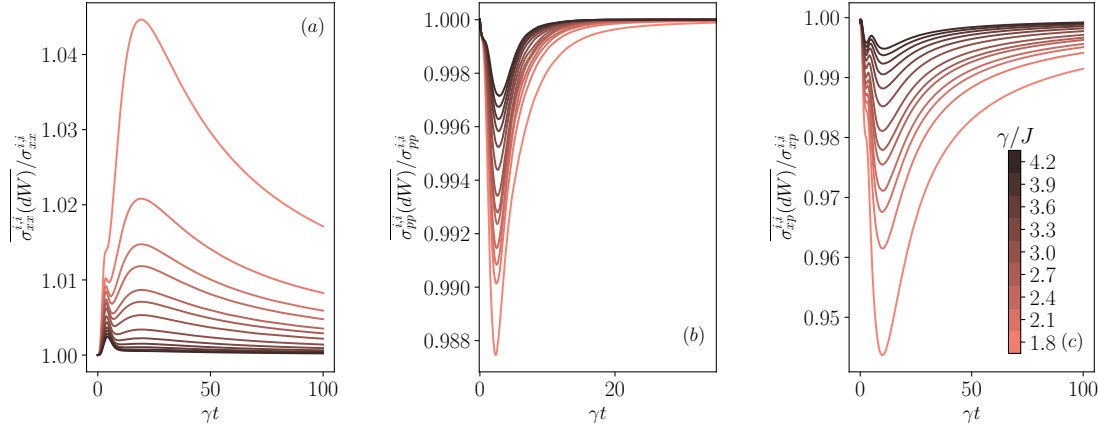

Figure 4: Ratio between trajectory averaged results obtained from quantum state diffusion and 1000 trajectories, and the results along the most-likely trajectory for (a) $\sigma_{xx}^{i,i}$, (b) $\sigma_{pp}^{i,i}$, (c) $\sigma_{xp}^{i,i}$. The parameters for the plot are $N = 7$, $\omega/J = 1/2$, $\alpha = 2.1$. The legend is shared among the panels.

dependence of these objects for simplicity. The equations read

$$\partial_t \langle \hat{x}_i \rangle = \omega \langle \hat{p}_i \rangle, \tag{37a}$$

$$\partial_t \langle \hat{p}_i \rangle = -g_i(t) - 2(\omega + h_i(t)) \langle \hat{x}_i \rangle + \omega (\langle \hat{x}_{i+1} \rangle + \langle \hat{x}_{i-1} \rangle), \tag{37b}$$

$$\partial_t \sigma_{xx}^{i,j} = \omega \left( \sigma_{xp} + \sigma_{px} \right)^{ij} - 4\gamma (\sigma_{xp} \sigma_{xp}^T)^{ij} + \gamma \delta_{ij}, \tag{37c}$$

$$\partial_t \sigma_{pp}^{i,j} = \left[ -\omega(2 + r_N^2)\sigma_{xp}^{i,j} + \omega(\sigma_{xp}^{i-1,j} + \sigma_{xp}^{i+1,j}) - 2\sigma_{xp}^{i,j} h_j(t) + (i \leftrightarrow j) \right] - 4\gamma (\sigma_{pp}^2)^{ij}, \tag{37d}$$

$$\partial_t \sigma_{xp}^{i,j} = \omega(\sigma_{pp}^{i,j} - (2 + r_N^2)\sigma_{xx}^{i,j} + \sigma_{xx}^{i,j-1} + \sigma_{xx}^{i,j+1}) - 2h_j(t)\sigma_{xx}^{i,j} - 2\gamma((\sigma_{xp} + \sigma_{px})\sigma_{pp})^{ij}. \tag{37e}$$

Despite the deterministic equations being *formally* the same as (36), there is a fundamental difference as $\langle \hat{x}_i \rangle$ is now deterministic. This makes the effective mass $h_i(t)$ and drive $g_i(t)$ along with all the correlations $\boldsymbol{\sigma}_{xx}, \boldsymbol{\sigma}_{xp}, \boldsymbol{\sigma}_{pp}$ deterministic in turn. The closed set of equations (37) can now be solved analytically, and here lies the power of our approach.

However, it should be noticed that in contrast with the free case, equations (37) do not correspond to the averaged stochastic equations (36), due to the more complex structure in the stochastic equations as

$$\overline{\sigma_{ab}^{i,j} h_i(t; dW)} \neq \sigma_{a,b}^{ij} h_i(t; 0), \text{ with } a, b = x \text{ or } p. \tag{38}$$

Nonetheless, the most likely trajectory approximation remains reliable, as can be checked in Fig. 4, which reports the comparison between trajectory-averaged correlations and correlations along the most-likely trajectory along the time evolution. From these plots, it is clear that the most likely trajectory approach describes with great accuracy the steady-state of our system within the SCTDHA.

## 4.3 Steady state properties

We will now use most-likely trajectory approach within the SCTDHA to discuss the properties of the steady state of the Sine Gordon model. We highlight once more the power of the most-likely trajectory approach by noticing that it allows to determine the steady state analytically

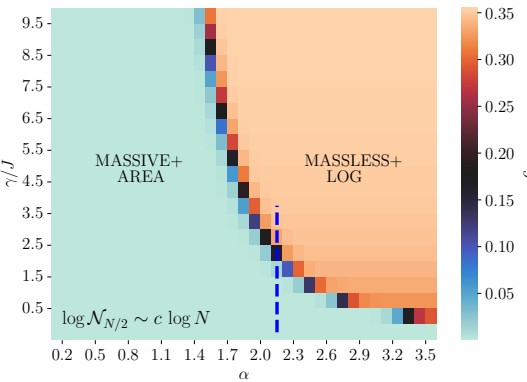

Figure 5: Phase diagram of the model: the color code represents the fitting parameter $c$ such that $\log \mathcal{N} \sim c \log N$ for $\omega/J = 1/2$. The plot shows the phase transition between the area-law massive region and the log-law massless region which reproduces the Free Bosons CFT results. The dashed line corresponds to the values of $\gamma/J$ and $\alpha/J$ used in Figs. 4 and 6. The plot is obtained for $\omega/J = 1/2$.

from Eq.s (37c)-(37e) for any system size, in sharp contrast with the stochastic state diffusion approach which would require (i) the generation of a full ensemble of trajectories, (ii) the evolution of the correlation matrix from $t = 0$ to the steady-state for each trajectory, (iii) averaging over all the trajectory evolutions and studying the steady state properties.

The steady-state equations for determining the correlations can be obtained through a Fourier transform exploiting translation invariance. To simplify the equations we set, without loss of generality, the initial state to have $\langle \hat{x}_j \rangle = \langle \hat{p}_j \rangle = 0 \forall j$.

$$\sigma_{xp}^q = \sigma_{xp} = \frac{\omega - \sqrt{\omega^2 + 4\gamma^2}}{4\gamma}, \tag{39}$$

$$\sigma_{pp}^q = \sqrt{-\frac{\omega}{2\gamma}\sigma_{xp}} \sqrt{4\sin\left(\frac{q}{2}\right)^2 + r_N^2 + \frac{J}{\omega}e^{-\frac{\alpha^2}{2}S_x}}, \tag{40}$$

$$\sigma_{xx}^q = \sqrt{-\frac{\omega}{2\gamma}\sigma_{xp}\left(1 - \frac{4\gamma}{\omega}\sigma_{xp}\right)} \frac{1}{\sqrt{4\sin\left(\frac{q}{2}\right)^2 + r_N^2 + \frac{J}{\omega}e^{-\frac{\alpha^2}{2}S_x}}}, \tag{41}$$

$$S_x = \frac{1}{N}\sum_q \sigma_{xx}^q \sim \sqrt{-\frac{\omega}{2\gamma}\sigma_{xp}\left(1 - \frac{4\gamma}{\omega}\sigma_{xp}\right)} \frac{4}{\pi} \frac{1}{\sqrt{r_N^2 + \frac{J}{\omega}e^{-\frac{\alpha^2}{2}S_x} + 4}}.$$
$$\cdot \, \mathrm{K}\left(\frac{4}{4 + r_N^2 + \frac{J}{\omega}e^{-\frac{\alpha^2}{2}S_x}}\right), \tag{42}$$

where the last equation is needed for self-consistency, and $\mathrm{K}(\bullet)$ is the complete elliptic integral of the first kind. In particular, $S_x$ defines the steady-state value of the effective mass:

$$h_j = h = \frac{J}{2}e^{-\frac{\alpha^2}{2}S_x}. \tag{43}$$

The study of the results of the steady-state equations gives rise to the phase diagram in Fig. 5. The phase diagram shows the presence of two regions: a massive, area-law phase found for $\alpha$ and $\gamma/J$ small, and a massless, log-law phase found in the opposite regime.

The mass properties are obtained from the analysis of the effective mass from the SCT-DHA, $h_j$, while the entanglement properties are obtained from the analysis of the half-chain

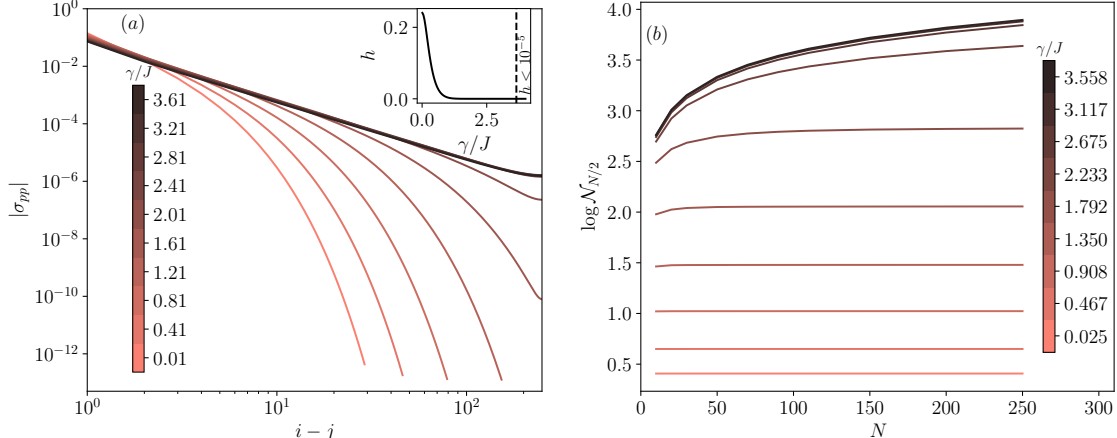

Figure 6: Some steady-state properties for the Sine-Gordon model with $\omega/J = 1/2$. (a) Momentum-momentum correlations in the main plot for $N = 500$ and fixed $\alpha = 2.1$. The plot is in log-log scale, showing that when the model is massless ($\gamma/J \gtrsim 2.64$), correlations decrease polynomially, in contrast with lines corresponding to a massive model, which decrease exponentially. Inset: behaviour of the effective mass vs $\gamma/J$ for $\alpha = 2.1$. The vertical dashed line signals $\gamma/J \gtrsim 2.64$, beyond which $h < 10^{-5}$. (b) Logarithmic Negativity for fixed $\alpha = 2.1$ and $\omega/J = 1/2$ for several measurement strengths, showing the change from area law to log law by increasing the measurement strength.

logarithmic negativity. Notice how the premises about Gaussian fluctuations introduced in Eq. (27), highlight the accuracy of the phase diagram in the strong-measurements limit.

Analyzing the phase diagram at fixed $\alpha$, for instance by setting the value indicated by the blue dashed line in Fig. 5, one can notice interesting properties of the model. Consistent with the localization-delocalization picture proposed at the beginning, we notice that at fixed $\alpha$, the massive region is found for low $\gamma/J$. This indicates that the Sine-Gordon potential wins the competition with the weak measurements and confines the $j^{\text{th}}$ oscillator in one of the wells of the cosine. This is expressed by the finite mass of the SCTDHA as the $h$ coefficient is related to the second-order expansion at the bottom of the wells of the cosine. In contrast, for higher values of $\gamma/J$, the stronger measurements do not allow the oscillator to localize in a well of the Sine-Gordon potential, resulting in $h = 0$ as localization is not possible anymore.

The massive-massless transition, which is how the SCTDHA interprets the delocalization process arising from momentum measurements, is signaled not only by changes in the value of the effective mass $h$, but also by the transition between exponentially decaying to power-law decaying momentum-momentum correlations, and also by the transition between area law and log-law scalings in the logarithmic negativity. This is shown in Fig. 6, displaying for $\alpha = 2.1$ (corresponding to the dashed line in Fig. 5) both the momentum-momentum correlation properties, panel (a), and entanglement properties, panel (b).

Some comments are in order. The results we obtain for $\alpha \gg 1$ are consistent with the Free Bosons CFT's results predicted in Ref. [49]. We obtain, as expected, a massless theory with power-law decaying correlations and a logarithmic entanglement. Moreover, the phase diagram in Fig. 5 shows only a finite set of $\gamma/J$ values. Increasing them one does not find a saturation value for the critical value of $\alpha$ ($\alpha_C$) which signals the transitions, as the steady state equations indicate an increasing shrinking of the massive phase with $\alpha_C \to 0$. Thus, projective measurement would inevitably lead to a total delocalization of the bosons, yielding a massless phase only.

The most-likely trajectory approach within the SCTDHA is thus predicting some striking results for the Sine-Gordon Model: for fixed $\alpha$, measurements are tuning a transition from a massive, localized, area-law region to a massless, delocalized, logarithmic region comprising the Free Bosons CFT behaviour. The competition between the Sine-Gordon potential and the momentum measurements is, as anticipated, driving a non-trivial change in the behaviour of the system.

## 4.4 Perturbation theory

The measurement-induced phase transition we have found for the Sine-Gordon monitored theory has been obtained following the most likely trajectory approach within the self-consistent time-dependent harmonic approximation. In particular, the combination of these two methods has allowed us to derive analytical equations to witness the emergence of the monitored criticality. While, as proved in Figs. 4, the most-likely trajectory is reliable and its prediction can be trusted, it is important to explore the validity of the SCTDHA to confirm the predictions made in this Section.

The SCTDHA has been tested for the Sine-Gordon model at equilibrium and in response to quenches, showing a good agreement with exact results [50, 51]. As monitoring goes beyond these settings, the aim is now to understand whether and where the cosine operator is relevant for the monitored theory, indicating the appearance of a phase transition. This would corroborate the SCTDHA predictions for the monitored Sine-Gordon theory, especially for the MIPT found in the previous Section.

In order to achieve this, we study the monitored Sine-Gordon model in Eq.(33) in perturbation theory for $J \gg 1$, with $J$ being the strength of the Sine-Gordon potential. This yields as zeroth-order theory the monitored Free Bosons CFT studied in Sec. 3.

The object we are interested in calculating is

$$I_{i,j} = \left\langle \cos(\alpha \, \hat{x}_i) \cos(\alpha \, \hat{x}_j) \right\rangle_0, \tag{44}$$

where $\langle \bullet \rangle_0$ indicates quantum averages over the steady state of the free bosons CFT with momentum measurements. This object is relevant for our purposes as it accounts for both the first-order corrections to the average value of the cosine operator and for zeroth-order cosine-cosine correlations.

The cosine-cosine correlations in Eq. (44) can be calculated by exploiting the Gaussian nature of the monitored free bosons theory. Introducing the sum and difference operators $\hat{\eta}_{i,j} = \hat{x}_i + \hat{x}_j$, $\hat{\varepsilon}_{i,j} = \hat{x}_i - \hat{x}_j$, which follow a Gaussian theory in turn, we can rewrite

$$
\begin{aligned}
I_{i,j} &= \frac{1}{2} \left( \left\langle \cos(\alpha \, \hat{\eta}_{i,j}) \right\rangle_0 + \left\langle \cos(\alpha \, \hat{\varepsilon}_{i,j}) \right\rangle_0 \right) \\
&= \frac{1}{2} \left( \cos(\alpha \left\langle \hat{\eta}_{i,j} \right\rangle_0) e^{-\frac{1}{2}\alpha^2 \sigma_{\eta_{ij}}} + \cos(\alpha \left\langle \hat{\varepsilon}_{i,j} \right\rangle_0) e^{-\frac{1}{2}\alpha^2 \sigma_{\varepsilon_{ij}}} \right),
\end{aligned}
\tag{45}
$$

where we have introduced the variances of the sum and difference operators: $\sigma_{\eta_{ij}}, \sigma_{\varepsilon_{ij}}$. Notice that Eq. (45) contains, both in the sum and the difference term, two important factors: a cosine factor and an exponential factor. The cosine factors $\cos(\alpha \left\langle \hat{\eta}_{i,j} \right\rangle_0)$ and $\cos(\alpha \left\langle \hat{\varepsilon}_{i,j} \right\rangle_0)$ are stochastic when computed along specific trajectories, as they depend only on averages of $\hat{x}_i$ (see App. B). Moreover, exploiting translational invariance, the cosine factors do not explicitly depend on the $i, j$ indices. The exponential factors, instead, are purely deterministic as the variance equations for the monitored free bosons CFT do not contain stochastic terms.

Let us evaluate the deterministic terms first. The arguments of the two exponentials are

$$\sigma_{\eta_{ij}} = \sigma_{xx}^{ii} + \sigma_{xx}^{jj} + 2\sigma_{xx}^{ij} = \frac{2}{N}\sum_q (1 + \cos(q(i-j)))\,\sigma_{xx}^q, \tag{46}$$

$$\sigma_{\varepsilon_{ij}} = \sigma_{xx}^{ii} + \sigma_{xx}^{jj} - 2\sigma_{xx}^{ij} = \frac{2}{N}\sum_q (1 - \cos(q(i-j)))\,\sigma_{xx}^q, \tag{47}$$

where $\sigma_{xx}$ in the steady state can be deduced from Eq. (41) imposing $J = 0$. Notice that the equations for $\sigma_{xx}$ are the same for the stochastic and most likely theories. Evaluating these Fourier transforms in the thermodynamic limit and continuous $y = i - j \in \mathbb{R}$ limit, we obtain

$$\frac{2}{N}\sum_q (1 \pm \cos(q\,y))\,\sigma_{xx}^q \sim \frac{\Gamma}{\pi}\int dq\,\frac{1 \pm \cos(q\,y)}{|q|}e^{-q\Lambda} = \begin{cases} +\infty & \text{for } \sigma_{\eta_{ij}} \\ \frac{\Gamma}{\pi}\log\left(1 + \frac{y^2}{\Lambda^2}\right) & \text{for } \sigma_{\varepsilon_{ij}} \end{cases}, \tag{48}$$

where $\Lambda$ is a UV cutoff and $\Gamma = \sqrt{-\frac{\omega}{2\gamma}\sigma_{xp}(1 - \frac{4\gamma}{\omega}\sigma_{xp})}$ with $\sigma_{xp}$ is the steady-state value for the $\sigma_{xp} = \frac{\omega - \sqrt{\omega^2 + 4\gamma^2}}{4\gamma}$.

Thus, the cosine-cosine correlations scale as

$$I(y) = \frac{1}{2}\cos\alpha\,\langle\hat{\epsilon}_y\rangle_0\left(\frac{\Lambda^2}{\Lambda^2 + y^2}\right)^{\frac{\alpha^2\Gamma}{2\pi}}. \tag{49}$$

Since translational invariance causes $\langle\hat{\epsilon}_y\rangle$ not to depend on $y$, the scaling of correlations is entirely determined by the deterministic terms. In particular, for each trajectory, the steady-state contribution of the cosine operator in the action is determined by the convergence of the integral $\int dy\,\sqrt{I(y)}$ [61, 62]. Namely:

$$\int dy\,\left(\frac{\Lambda^2}{\Lambda^2 + y^2}\right)^{\frac{\alpha^2\Gamma}{4\pi}} < \infty \iff \frac{\alpha^2\Gamma}{2\pi} > 1. \tag{50}$$

Thus, all the trajectories, including the most likely one, share the same critical line for the cosine operator to be relevant. This means that this can be identified as the critical line for the monitored theory and proves the goodness of the SCTDHA approximation for our monitored theory. The comparison with the results obtained in the previous section are shown in Fig.7(a). The SCTDHA and perturbation theory critical lines have an $O(1)$ ratio, proving a qualitative agreement with the results from the SCTDHA and highlighting the valifity of the latter for describing the monitored steady state of the Sine-Gordon model.

Interestingly, averaging over trajectories one obtains that the cosine operator is never relevant as $\overline{I(y)} \propto \overline{\cos(\alpha\langle\hat{\epsilon}_y\rangle)} = 0$ in the steady state, as shown numerically in Fig.7(b) This is consistent with the results that one would obtain by performing the same calculation as in this section using the $\sigma_x^q$ obtained from the Lindblad equation, which diverges with time due to decoherence, and highlight once again the measurement-induced nature of the phase transitions we see emerging.

## 5  Discussion and Conclusions

In this paper, we have developed a new method to address the monitored dynamics of many-body bosonic systems. The method is based on the concept of the most-likely trajectory, demonstrating how the trajectory that dominates the joint probability distribution of quantum trajectories can effectively describe the full monitored dynamics alone. We have benchmarked

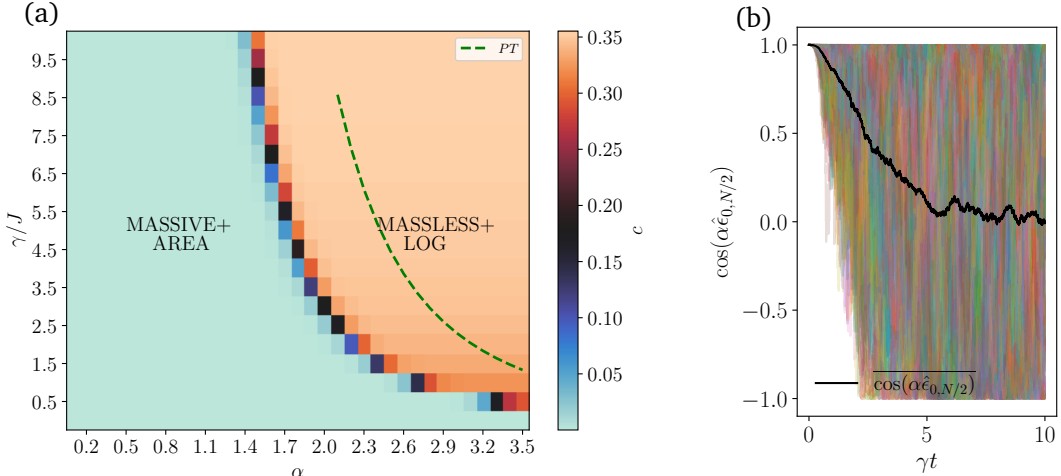

Figure 7: (a) Comparison between theoretical results obtained in perturbation theory (green dashed line), and results obtained in the SCTDHA. The critical lines show qualitatively the same behaviour, confirming the validity of the SCTDHA for the monitored Sine-Gordon model (b) Average over 150 trajectories of the cosine of the difference operator $\hat{\varepsilon}$ (black line), calculated in this case for $i = 0, j = N/2$. Due to translational invariance, the same plot would have been obtained for other choices. The opaque colored lines in the background correspond to the trajectory-results.

our method with free boson CFT, exactly reproducing the results obtained in Ref. [49]. The validity of our method in free theories has motivated us to explore its application in interacting systems, which were previously inaccessible analytically because of the complications emerging from interactions and the stochastic nature of the measurement process. Specifically, following our most likely trajectory method to the Self-Consistent Time-Dependent Harmonic Approximation of the Sine-Gordon model has allowed us to close the equations of motion at the level of second moments, and to study the dynamics along a single trajectory. This has enabled us to identify deterministically the steady state of the monitored dynamics and to reliably identify a measurement-induced phase transition in the logarithmic negativity, signaling a change from an area-law to a logarithmic scaling of entanglement.

Our approach has thus proven capable not only of exactly recovering known results in free theories but also of revealing new phenomena emerging in interacting systems that were previously inaccessible.

The strength of the method lies in its ability to produce deterministic equations of motion that are possible to solve analytically, significantly simplifying the study of monitored quantum systems. While we have applied this approach to bosonic systems, it would be important to test its applicability in models of a different nature, such as fermionic or spin systems. In particular, we expect promising results for semiclassical spin systems, which can often be effectively bosonized.

It would be interesting as well to connect our method to other well-known types of non-Hermitian deterministic master equations, such as the no-click or partial monitoring.

# Acknowledgements

We would like to thank M. Schirò and M. Buchhold for helpful discussions.

**Funding information**   This work was supported by PNRR MUR project PE0000023- NQSTI, by the European Union (ERC, RAVE, 101053159). Views and opinions expressed are however those of the author(s) only and do not necessarily reflect those of the European Union or the European Research Council. Neither the European Union nor the granting authority can be held responsible for them.

## A   Path Integrals with Continuous Measurements

In this Appendix, we will derive the path-integral expression of the joint probability distribution of a quantum trajectory. We can start by expanding Eq. (5) for position measurements with an initial state $\hat{\rho}(0) = \int dx_0 dx'_0 \, \rho(x_0, x'_0) \, |x_0\rangle \langle x'_0|$, considering for the derivation just a single-body framework:

$$P(r; t) = \lim_{\substack{\delta t \to 0 \\ K \to \infty}} \mathrm{Tr}\{\hat{U}_{\delta t} \hat{M}_{r_K} \hat{U}_{\delta t} \hat{M}_{r_{K-1}} ... \hat{U}_{\delta t} \hat{M}_{r_1} \hat{\rho}(0) \hat{M}_{r_1} \hat{\mathcal{U}}_{\delta t}^{\dagger} ... \hat{M}_{r_{K-1}} \hat{\mathcal{U}}_{\delta t}^{\dagger} \hat{M}_{r_K} \hat{\mathcal{U}}_{\delta t}^{\dagger}\} \quad (A.1)$$

$$= \lim_{\substack{\delta t \to 0 \\ K \to \infty}} \int dx_0 dx'_0 \, dx \, \rho(x_0, x'_0) \, \langle x| \hat{U}_{\delta t} \hat{M}_{r_K} \hat{U}_{\delta t} \hat{M}_{r_{K-1}} ... \hat{U}_{\delta t} \hat{M}_{r_1} |x_0\rangle \cdot \quad (A.2)$$

$$\cdot \langle x'_0| \hat{M}_{r_1} \hat{\mathcal{U}}_{\delta t}^{\dagger} ... \hat{M}_{r_{K-1}} \hat{\mathcal{U}}_{\delta t}^{\dagger} \hat{M}_{r_K} \hat{\mathcal{U}}_{\delta t}^{\dagger} |x\rangle$$

$$= \int dx_0 dx'_0 \, dx \, \rho(x_0, x'_0) \mathcal{K}(x, t; x_0, 0) \mathcal{K}^*(x, t; x'_0, 0) \quad (A.3)$$

In order to build the propagator $\mathcal{K}(x, t; x_0 0)$ we can exploit that, according to Hausdorff-Campbell-Baker formula, in the limit $\delta t \to 0$ we can approximate

$$e^{-i\delta t \hat{H}} e^{-i\gamma \delta t (r - \hat{x})^2} \sim e^{-i\delta t \hat{H}_{\mathrm{nh}}} = e^{-i\delta t (\hat{H} - i\gamma (r - \hat{x})^2)}. \quad (A.4)$$

We can thus build the path integral propagator $\mathcal{K}$ à la Feynman using the non-Hermitian effective Hamiltonian $\hat{H}_{\mathrm{nh}}$, getting

$$\mathcal{K}(x, t; x_0, 0) = \int_{x(0)=x_0}^{x(t)=x} \mathcal{D}x_1(t') \exp\{iS_{\mathrm{nh}}\} \quad (A.5)$$

$$S_{\mathrm{nh}} = S_0[x_1; t] + i\gamma \int_0^t dt' \big(r(t') - x_1(t')\big)^2 \quad (A.6)$$

with $S_0$ being the action related to the Hamiltonian $\hat{H}$.

The same thing can be done to obtain $\mathcal{K}^*(x, t; x'_0, 0)$ taking care of the fact that the unitary evolution operator is non-Hermitian, while the measurement operator is in our case Hermitian.

$$\mathcal{K}^*(x, t; x_0, 0) = \int_{x(0)=x'_0}^{x(t)=x} \mathcal{D}x_2(t') \exp\{-iS_{\mathrm{nh}'}\} \quad (A.7)$$

$$S_{\mathrm{nh}'} = S_0[x_2; t] - i\gamma \int_0^t dt' \big(r(t') - x_2(t')\big)^2. \quad (A.8)$$

Notice the change in the sign of the measurement term.

The fields $x_1(t)$ and $x_2(t)$ correspond to the forward and backward path fields in Keldysh theory, and putting $\mathcal{K}$ and $\mathcal{K}^*$ one gets Eq. (8).

### A.1 Gaussian Fluctuations

We can now use this formalism to evaluate Gaussian fluctuations around the Saddle Point (most-likely) trajectory.

Consider the following setting (without loss of generality): we study the monitored evolution of a many-body system on an $N$ sites lattice, with Hamiltonian $\hat{H}_0$ (leading to the action $S_0$), subject to $\hat{R}_i$ measurements on each site, $i = 1, .., N$. We will thus use the measurement operator with the same structure as in Eq. (29).

Introducing the N-components classical and quantum Keldysh fields

$\mathbf{R}_+(t') = (R_{+,1}(t'), ..., R_{+,N}(t')), \mathbf{R}_-(t') = (R_{-,1}(t'), ..., R_{-,N}(t'))$ with $R_{+,i} = \frac{R_{1,i}+R_{2,i}}{2}, R_{-,i} = R_{1,i} - R_{2,i}$,

the average trajectory can be calculated in representation as

$$\overline{r_j(t)} = \int \mathcal{D}\mathbf{r}(t')\, r_j(t)\, e^{\mathcal{S}[\mathbf{r};t]} \tag{A.9}$$

$$\mathcal{S}[\mathbf{r};t] = \log(P[\mathbf{r};t]) \tag{A.10}$$

$$= \log\left( \int d\mathbf{R}_0\, d\mathbf{R}_0'\, d\mathbf{R}\, \rho(\mathbf{R}_0 + \frac{\mathbf{R}_0'}{2}, \mathbf{R}_0 - \frac{\mathbf{R}_0'}{2}) \int_{\mathbf{R}_0}^{\mathbf{R}} \mathcal{D}\mathbf{R}_+(t') \int_{\mathbf{R}_0'}^{0} \mathcal{D}\mathbf{R}_-(t') \cdot \right.$$

$$\left. \cdot\, e^{iS_0[\mathbf{R}_+,\mathbf{R}_-] - \gamma \sum_{j=1}^{N} \int_0^t dt' \left[ 2\left(r_j(t') - R_{+,j}(t')\right)^2 + \frac{1}{\gamma} R_{-,j}(t')^2 \right]} \right)$$

The $j^{\text{th}}$ Saddle Point trajectory can be found by solving the equation

$$\left. \frac{\partial \mathcal{S}[\mathbf{r};t]}{\partial r_j(\tau)} \right|_{r_j^*(\tau)} = 0$$

$$\Rightarrow \quad \frac{-4\gamma}{P[\mathbf{r}^*;t]} \left( r_j^*(\tau) P[\mathbf{r}^*;t] - \left\langle \hat{R}_{+,j}(\tau) \right\rangle_{\tilde{\hat{\rho}}_{r^*}} \right) = 0 \tag{A.11}$$

where $\tilde{\hat{\rho}}_{\mathbf{r}^*(\tau)}$ is the un-normalized density matrix evolved along $\mathbf{r}^*$ up to time $\tau$- In particular, we find

$$r_j^*(\tau) = \left\langle \hat{R}_j \right\rangle_{\hat{\rho}_{r^*(\tau)}} \tag{A.12}$$

For Gaussian systems, for which the Saddle point is exact, we find $\overline{\mathbf{r}(t)} = \mathbf{r}^*(t)$, indicating that the trajectory defined in (13) maximizes both the joint and the conditional probability distribution.

We can now evaluate the second-order expansion around the $r^*$ trajectory:

$$\mathcal{S}[\mathbf{r};t] \sim \mathcal{S}[\mathbf{r}^*;t] + \frac{1}{2} \int_0^t dt'dt'' \left(\mathbf{r}^*(t') - \mathbf{r}(t')\right) \mathbb{S}^{(2)}[\mathbf{r}^*;t',t''] \left(\mathbf{r}^*(t'') - \mathbf{r}(t'')\right). \tag{A.13}$$

The second-order Kernel can be obtained by deriving (A.11) once again:

$$\left. \frac{\partial^2 \mathcal{S}[\mathbf{r};t]}{\partial r_j(t')\partial r_i(t'')} \right|_{\mathbf{r}^*} = \tag{A.14}$$

$$= \frac{4\gamma}{P[\mathbf{r}^*;t]^2} \left(r_j^*(t')P[\mathbf{r}^*,t] + \left\langle\hat{R}_{+,j}(t')\right\rangle_{\tilde{\hat{\rho}}_{\mathbf{r}^*}}\right)\left(r_i^*(t'')P[\mathbf{r}^*,t] + \left\langle\hat{R}_{+,i}(t'')\right\rangle_{\tilde{\hat{\rho}}_{\mathbf{r}^*}}\right) - 4\gamma\delta_{i,j}\delta(t'-t'') +$$

$$+ \frac{16\gamma^2}{P[\mathbf{r}^*;t]} \left(r_j^*(t')r_i^*(t'')P[\mathbf{r}^*;t] - r_j^*(t)\left\langle\hat{R}_{+,i}(t'')\right\rangle_{\hat{\rho}_{\mathbf{r}^*}} - r_i^*(t'')\left\langle\hat{R}_{+,j}(t')\right\rangle_{\hat{\rho}_{\mathbf{r}^*}} + \left\langle\hat{R}_{+,i}(t'')\hat{R}_{+,j}(t')\right\rangle_{\hat{\rho}_{\mathbf{r}^*}}\right)$$

$$= -4\gamma\delta_{i,j}\delta(t'-t'') + 16\gamma^2\left[\left\langle\hat{R}_{+,j}(t')\hat{R}_{+,i}(t'')\right\rangle_{\hat{\rho}_{\mathbf{r}^*}} - \left\langle\hat{R}_{+,j}(t')\right\rangle_{\hat{\rho}_{\mathbf{r}^*}}\left\langle\hat{R}_{+,i}(t'')\right\rangle_{\hat{\rho}_{\mathbf{r}^*}}\right]$$

590     Using the brief label

$$\sigma_{R_iR_j}^{++}(t',t'') = \left\langle \hat{R}_{+,j}(t')\hat{R}_{+,i}(t'') \right\rangle_{\hat{\rho}_{\mathbf{r}^*}} - \left\langle \hat{R}_{+,j}(t') \right\rangle_{\hat{\rho}_{\mathbf{r}^*}} \left\langle \hat{R}_{+,i}(t'') \right\rangle_{\hat{\rho}_{\mathbf{r}^*}},$$

591     we get

$$\mathbb{S}_{ij}^{(2)}[\mathbf{r}^*;t',t''] = -4\gamma\delta_{i,j}\delta(t'-t'') + 16\gamma^2\sigma_{R_iR_j}^{++}(t',t''). \tag{A.15}$$

592     While being exact for quadratic theories like free bosons, this result highlights how the
593 most-likely trajectory approach is reliable for interacting theories, depending on the measure-
594 ment strength and the classical-field correlations of the operator we are measuring.

## 595   B   Exact Results for the Free Bosons CFT

596     In this appendix, we report results obtained from the Free Bosons CFT model using a quantum
597 state diffusion perspective. This approach gives exact results when averaging over trajectories.
598     The Hamiltonian and the measurements we consider are contained in Eqs. (28), (29). The
599 master equation describing the evolution along a $dW$ stochastic trajectory is:

$$\frac{\partial\hat{\rho}(t)}{\partial t} = -i\left[\hat{H}_{\mathrm{FB}},\hat{\rho}(t)\right] + \gamma\sum_{j=1}^{N}\left(\hat{p}_j\hat{\rho}(t)\hat{p}_j - \frac{1}{2}\left\{\hat{p}_j^2,\hat{\rho}(t)\right\}\right) \tag{B.1}$$

$$+ \sqrt{\gamma}\sum_{j=1}^{N}dW_j\left\{\hat{p}_j - \left\langle\hat{p}_j\right\rangle_t,\hat{\rho}(t)\right\}$$

600 .

601     Exploiting the Gaussianity for the model, we can write the Heisenberg equations to evolve
602 averages and correlations of $\hat{x}_i, \hat{p}_j$, which close at the level of second-order moments.

$$\partial_t\left\langle\hat{x}_i\right\rangle = \omega\left\langle\hat{p}_i\right\rangle + 2\sqrt{\gamma}\sum_{j=1}^{N}dW_j\,\sigma_{xp}^{ji} \tag{B.2a}$$

$$\partial_t\left\langle\hat{p}_i\right\rangle = -2\omega\left\langle\hat{x}_i\right\rangle + \omega\left(\left\langle\hat{x}_{i+1}\right\rangle + \left\langle\hat{x}_{i-1}\right\rangle\right) + 2\sqrt{\gamma}\sum_{j=2}^{N}dW_j\,\sigma_{pp}^{ij} \tag{B.2b}$$

$$\partial_t\sigma_{xx}^{i,j} = \omega\left(\sigma_{xp}+\sigma_{px}\right)^{ij} - 4\gamma(\sigma_{xp}\sigma_{xp}^T)^{ij} + \gamma\delta_{ij} \tag{B.2c}$$

$$\partial_t\sigma_{pp}^{i,j} = \omega\left[-(1+\frac{r_N^2}{2})\sigma_{xp}^{i,j} + (\sigma_{xp}^{i-1,j}+\sigma_{xp}^{i+1,j}) + (i\leftrightarrow j)\right] - 4\gamma(\sigma_{pp}^2)^{ij} \tag{B.2d}$$

$$\partial_t\sigma_{xp}^{i,j} = \omega(\sigma_{pp}^{i,j} - (2+r_N^2)\sigma_{xx}^{i,j} + \sigma_{xx}^{i,j-1} + \sigma_{xx}^{i,j+1}) - 2\gamma((\sigma_{xp}+\sigma_{px})\sigma_{pp})^{ij} \tag{B.2e}$$

603     The Heisenberg equations show that the monitored dynamics along specific trajectories
604 obey stochastic trajectories for averages and deterministic equations for correlations. This is a
605 peculiarity of free systems, yielding a simple calculation for trajectory-averaged quantities. For
606 instance, the logarithmic negativity is determined only in terms of connected correlations and,
607 being deterministic in turn, is straightforward to average over trajectories. Indeed, exploiting
608 the properties of a Gaussian states [56–58]

$$\log\mathcal{N}_{N/2} = \log\mathrm{Tr}\,\hat{\rho}^{T_{N/2}} = \sum_n\log\max\left\{1,\frac{1}{2\nu_{\mathrm{n}}}\right\}, \tag{B.3}$$

609 where $\hat{\rho}^{T_{N/2}}$ is the partial transpose of a bipartition of the many-body density matrix, and $\nu_n$
610 are the symplectic eigenvalues of the partial transposed covariance matrix.

611     The covariance matrix can be written as

$$\boldsymbol{\sigma} = \begin{pmatrix} \boldsymbol{\sigma}_{xx} & \boldsymbol{\sigma}_{xp} \\ \boldsymbol{\sigma}_{px} & \boldsymbol{\sigma}_{pp} \end{pmatrix} \tag{B.4}$$

612 This is essentially a covariance matrix written on the $2N-$dimensional basis $\mathbf{r} = (x_1, ..., x_N, p_1, ..., p_N)$.
613 Its half-chain partial transpose can be obtained through an operator $\mathcal{T}_{N/2}$ as $\boldsymbol{\sigma}^{T_{N/2}} = \mathcal{T}_{N/2}\boldsymbol{\sigma}\mathcal{T}_{N/2}$,
614 where essentially $\mathcal{T}$ flips $\mathbf{r} = (x_1, ..., x_N, p_1, ..., p_{N/2}, ..., p_N) \rightarrow (x_1, ..., x_N, p_1, ..., -p_{N/2}, ..., -p_N)$.
615 The symplectic diagonalization is instead obtained by taking the non-negative eigen values of
616 $\boldsymbol{\sigma}' = i\boldsymbol{\Omega}\boldsymbol{\sigma}^{T_{N/2}}$, where

$$\boldsymbol{\Omega} = \begin{pmatrix} \mathbb{O}_N & \mathbb{I}_N \\ -\mathbb{I}_N & \mathbb{O}_N \end{pmatrix}, \tag{B.5}$$

617 and $\mathbb{O}_N$ is a $N \times N$ null matrix.

618     The half-chain logarithmic negativity for the monitored free bosons CFT has a logarithmic
619 behaviour

$$\log \mathcal{N}_{N/2} \sim c \log N \tag{B.6}$$

620 for any value of the measurement strength. Notice that for $\gamma \rightarrow 0$ we reproduce the known
621 results for free bosons having unity central charge [63, 64].

## C    The Self-Consistent-Time-Dependent Harmonic Approximation

623 In this Section, we report a detailed calculation to obtain the SCTDHA approximation [50, 51]
624 of the Hamiltonian of the Sine-Gordon model studied in the main text, expanding the steps
625 presented in the main text.

626     The SCTDHA consists, first of all, of a shift in the operators, defined at every time step:

$$\hat{\xi}_j = \hat{x}_j - \langle \hat{x}_j \rangle_t, \quad \text{with } \langle \hat{\xi}_j \rangle_t = 0 \tag{C.1}$$

627 We can now approximate the Hamiltonian in terms of first and second powers of the fluctuation
628 operator only:

$$\tilde{H}'_{\text{int}} \sim \sum_{j=1}^N \left( \mathcal{C}_j(t) + \hat{\mathcal{V}}_j(t)\hat{\xi}_j + \hat{\mathcal{M}}_j(t)\hat{\xi}_j^2 \right) \tag{C.2}$$

$$\hat{\mathcal{V}}_j(t) = \frac{\partial H_{\text{int}}\left( \hat{x}_j(t) \right)}{\partial \hat{x}_j(t)} \Bigg|_{\langle \hat{x}_j \rangle_t + \hat{\xi}_j(t)} \tag{C.3}$$

$$\hat{\mathcal{M}}_j(t) = \frac{1}{2} \frac{\partial^2 H_{\text{int}}\left( \hat{x}_j(t) \right)}{\partial \hat{x}_j(t)^2} \Bigg|_{\langle \hat{x}_j \rangle_t + \hat{\xi}_j(t)}, \tag{C.4}$$

629     and we perform the Hartree factorization: $\hat{\mathcal{V}}_j(t) \rightarrow \langle \hat{\mathcal{V}}_j(t) \rangle$, and $\hat{\mathcal{M}}_j(t) \rightarrow \langle \hat{\mathcal{M}}_j(t) \rangle$.
630 Specifically, we get for the first order parameter of the expansion:

$$\begin{aligned} \langle \hat{\mathcal{V}}_j(t) \rangle &= \frac{J}{\alpha} \left\langle \sin\left( \alpha \langle \hat{x}_j \rangle_t + \alpha \hat{\xi}_j \right) \right\rangle_t \\ &= \frac{J}{\alpha} \cos(\alpha \langle \hat{x}_j \rangle_t) \left\langle \sin(\alpha \hat{\xi}_j) \right\rangle_t + \frac{J}{\alpha} \sin(\alpha \langle \hat{x}_j \rangle_t) \left\langle \cos(\alpha \hat{\xi}_j) \right\rangle_t \\ &= \frac{J}{\alpha} e^{-\frac{\alpha^2}{2} \langle \hat{\xi}_j^2 \rangle_t} \sin(\alpha \langle \hat{x}_j \rangle_t). \end{aligned} \tag{C.5}$$

The second order is instead

$$
\begin{aligned}
\langle \hat{\mathcal{M}}_j(t) \rangle &= \frac{J}{2} \langle \cos\left( \alpha \langle \hat{x}_j \rangle_t + \alpha \hat{\xi}_j \right) \rangle_t && \text{(C.6)} \\
&= \frac{J}{2} \cos(\alpha \langle \hat{x}_j \rangle_t) \langle \cos(\alpha \hat{\xi}_j) \rangle_t - \frac{J}{2} \sin(\alpha \langle \hat{x}_j \rangle_t) \langle \sin(\alpha \hat{\xi}_j) \rangle_t \\
&= \frac{J}{2} e^{-\frac{\alpha^2}{2} \langle \hat{\xi}_j^2 \rangle_t} \cos(\alpha \langle \hat{x}_j \rangle_t)
\end{aligned}
$$

where the facts that the Hamiltonian is quadratic in $\hat{\xi}_j$ and $\langle \xi_j \rangle_t = 0 \forall t$ have been used to carry out the Gaussian averaged in the equations above. Inserting the relation $\hat{\xi}_j = \hat{x}_j - \langle \hat{x}_j \rangle_t$ and defining

$$
f_j = \mathcal{C}_j(t) + \langle \hat{x}_j \rangle_t \left( \langle \hat{\mathcal{M}}_j \rangle_t \langle \hat{x}_j \rangle_t - \langle \hat{\mathcal{V}}_j \rangle_t \right), \tag{C.7}
$$

$$
g_j = \langle \hat{\mathcal{V}}_j \rangle_t - 2 \langle \hat{\mathcal{M}}_j \rangle_t \langle \hat{x}_j \rangle_t, \tag{C.8}
$$

$$
h_j = \langle \hat{\mathcal{M}}_j \rangle_t, \tag{C.9}
$$

one finds the approximated Hamiltonian in Eq. (35).

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
