# Peer review of "Measurement-induced phase transition in interacting bosons from most likely quantum trajectory"

_SciPost Physics_

## Round 1 · Referee Report · Anonymous (Referee 1) · 2025-11-14

Strengths

1) Introduces a deterministic most-likely trajectory approach that is exact for free bosons and conceptually powerful.

2) Extends measurement-induced transition physics to an interacting bosonic field theory (Sine–Gordon), which is largely unexplored.

3) Provides nontrivial physical insight into entanglement scaling under continuous monitoring.

4) Methodologically relevant: avoids stochastic trajectory sampling and may be broadly applicable.

Weaknesses

1) For the interacting case, the most likely trajectory may not reveal the exact behavior and the results further rely on the SCTDHA approximation. This is fine but limitations may be more clearly discussed.

2) Some conceptual aspects (interpretation of most-likely trajectory, comparison to broader MIPT literature) could be clarified to increase accessibility.

Report

The manuscript presents a timely and conceptually appealing approach to monitored bosonic many-body dynamics based on the most-likely quantum trajectory. The authors show that this method is exact for free bosonic theories and combine it with the self-consistent time-dependent harmonic approximation to study the monitored Sine–Gordon model. The work uncovers a measurement-induced phase transition in a continuum interacting bosonic field theory, which is a significant and novel contribution to the rapidly developing field of measurement-induced phenomena. The method is technically sound within its assumptions, and the physical implications are interesting and relevant for both theory and experiment.

The main limitations lie in the interacting-case approximations, whose range of validity is not fully discussed, and in the need for clearer contextualization relative to existing MIPT literature.

Nevertheless, these issues are minor and do not detract from the value of the results. Overall, the manuscript is well suited for publication after minor clarifications that would strengthen its presentation and impact.

Requested changes

It would strengthen the manuscript if the authors provided a more quantitative discussion of the limitations of their approach. In particular, it would be valuable to address how deviations from the most-likely trajectory could manifest, and how such effects might be diagnosed within alternative frameworks, e.g., replica-based treatments or numerical trajectory simulations. Even if a full analysis is beyond scope, clarifying how one would detect or characterize such deviations would considerably improve the reader’s understanding of the method’s range of validity.

Recommendation

Publish (easily meets expectations and criteria for this Journal; among top 50%)

---

## Editorial Decision

in_refereeing